# Detection of Ovarian Cancer through Exhaled Breath by Electronic Nose: A Prospective Study

**DOI:** 10.3390/cancers12092408

**Published:** 2020-08-25

**Authors:** Francesco Raspagliesi, Giorgio Bogani, Simona Benedetti, Silvia Grassi, Stefano Ferla, Susanna Buratti

**Affiliations:** 1Department of Gynecologic Oncology, Fondazione IRCCS Istituto Nazionale dei Tumori di Milano, 20133 Milan, Italy; raspagliesi@istitutotumori.mi.it (F.R.); Stefano.ferla1@studenti.unimi.it (S.F.); 2Department of Food, Environmental and Nutritional Sciences (DeFENS), Università degli Studi di Milano, 20133 Milan, Italy; simona.benedetti@unimi.it (S.B.); silvia.grassi@unimi.it (S.G.); susanna.buratti@unimi.it (S.B.)

**Keywords:** ovarian cancer, early diagnosis, electronic nose, MOS sensors, K-NN models

## Abstract

Background: Diagnostic methods for the early identification of ovarian cancer (OC) represent an unmet clinical need, as no reliable diagnostic tools are available. Here, we tested the feasibility of electronic nose (e-nose), composed of ten metal oxide semiconductor (MOS) sensors, as a diagnostic tool for OC detection. Methods: Women with suspected ovarian masses and healthy subjects had volatile organic compounds analysis of the exhaled breath using e-nose. Results: E-nose analysis was performed on breath samples collected from 251 women divided into three groups: 86 OC cases, 51 benign masses, and 114 controls. Data collected were analyzed by Principal Component Analysis (PCA) and K-Nearest Neighbors’ algorithm (K-NN). A first 1-K-NN (cases vs. controls) model has been developed to discriminate between OC cases and controls; the model performance tested in the prediction gave 98% of sensitivity and 95% of specificity, when the strict class prediction was applied; a second 1-K-NN (cases vs. controls + benign) model was built by grouping the non-cancer groups (controls + benign), thus considering two classes, cases and controls + benign; the model performance in the prediction was of 89% for sensitivity and 86% for specificity when the strict class prediction was applied. Conclusions: Our preliminary results suggested the potential role of e-nose for the detection of OC. Further studies aiming to test the potential adoption of e-nose in the early diagnosis of OC are needed.

## 1. Introduction

It is estimated that over 22,000 new cases of ovarian cancer (OC) will be diagnosed and that 14,000 women will die of this disease in 2020 only in the U.S. OC is the fifth most common cancer in women and the most common cause of gynecologic cancer deaths, thus being one of the most lethal cancers in women [1]. Nearly three cases out of four were diagnosed in the advanced stage of the disease (International Federation of Gynecology and Obstetrics (FIGO) stage III or IV) at presentation [2]. The absence of valid screening programs and the rapid spread through the peritoneal surface represent the main determinants for ovarian cancer lethality. In fact, to date, no screening test is accurate and reliable enough to detect OC in women. Growing evidence suggested that clinical examination, transvaginal ultrasound, and CA-125 dosage are not enough to detect early-stage OC in the general population [3]. Recently, the Food and Drug Administration (FDA) has provided recommendations against the use of currently offered tests for OC screening, stating that these tests should not be used due to the high rate of false results [4].

Accumulating evidence has shown that analyzing the volatile organic compounds (VOC) in biological fluids such as blood, urine, and breath might provide promising biomarkers for several diseases, including solid tumors [5,6,7]. VOCs are gaseous molecules that can be sampled quickly and non-invasively from breath. They can originate either from within the body (endogenous VOCs) or from external sources, including medications and environmental exposure (exogenous VOCs). As other gasses (including O2 and CO2), VOCs pass from the bloodstream into the lungs. Nowadays, breath analyses are applied in medicine and clinical pathology as a non-invasive and comfortable way for individual health state evaluation [8,9,10,11]. Recently, e-nose has been used in breath analysis for the diagnosis of renal diseases, diabetes, asthma [8,9,10,11]. The application of the e-nose has extended in oncology, specifically in patients with lung cancer, and has also been investigated in the prostate, colonic, breast, and head and neck cancers with promising results [12,13,14,15,16,17,18,19]. Growing evidence suggests that cancer metabolism alters the volatile part of the metabolome that are present in various body fluids, including tissues, blood, and breath [20,21,22]. The central hypothesis of our work is that, since OC has characteristics of a systemic disease, it is possible that metabolomes produced by cancer cells, might be detected in the exhaled breath of OC patients. In the present paper, we aimed to collect preliminary data to determine whether this technology could be used as a diagnostic tool to triage patients with OC.

## 2. Results

Two hundred eighty-eight (288) exhaled breath samples were collected during the experimentation. Data from 37 patients were excluded since 28 samples were not analyzed on the same day of sample collection, and nine patients were excluded since they did not match inclusion criteria. Then, 251 subjects met the inclusion criteria and were included in the final analysis. The study population was divided into three main groups: (1) controls—healthy subjects without adnexal masses (114 patients; 45.4%); (2) OC cases—women with malignant adnexal masses (86 patients; 34.3%); (3) benign disease—women with suspect adnexal masses with evidence of benign disease at histological evaluation (51 patients; 20.3%). The main characteristics of the study population are displayed in Appendix A. Histopathological diagnosis of malignant masses included: high-grade serous ovarian cancer (*n* = 63), low-grade serous ovarian cancer (*n* = 8), mucinous ovarian cancer (*n* = 5), ovarian carcinosarcoma (*n* = 2), endometrioid ovarian cancer (*n* = 2), and other malignant entities (*n* = 6). The whole population of OC cases included 32 (37.2%) patients with early-stage disease (i.e., stage I–II), and 54 (62.8%) patients with advanced-stage OC. Details of the histopathological findings are reported in Appendix A.

### 2.1. E-Nose Analysis and Sensor Selection

Figure 1 shows the sensor signals collected during the sampling phase on a case (a) and a control (b) sample. The x-axis represents the sampling time, while the sensor responses are reported on the y-axis. As displayed in Figure 1, the sensor responses gradually increased during sampling, reaching a plateau after 15–20 s. From the histograms representing the responses collected after 40 s of sampling, it is possible to notice that the sensor signals were lower for the control sample, while for the OC case, three sensors (S2, S6, and S7) showed significantly increased signals. To select the MOS sensors relevant for OC cases detection, an interval plot was primarily built on the e-nose dataset transformed by column autoscaling to eliminate the numerical differences between variables. The interval plot shows the 95% confidence interval for the mean value of each group. The difference among groups is statistically relevant only if there is no overlap of the indicator bars, and the width of the intervals correlates with the amount of variation in the data.

Figure 2 shows the comparison performed on data collected by each e-nose sensor for two groups (OC cases, including early and advanced stage, and controls). A statistically significant difference between cases and controls was highlighted for eight sensors, while for two sensors (S4 and S10), the overlapping of the indicator bars showed that the two classes were not different at the given level of confidence (α: 0.05). Furthermore, controls were characterized by small intervals showing more consistent data and less variation, while the wide intervals of OC cases indicated a significantly higher variation.

A further Principal Component Analysis (PCA) elaboration was applied to data collected by the eight selected metal oxide semiconductor (MOS) sensors, to identify the most informative sensors useful in discriminating OC cases from non-cancer samples (i.e., controls and benign disease). Figure 3 displays the PCA score plots (a1 and a2) and loading plots (b) of the data collected on OC cases (red), benign disease (yellow), and controls’ (green) samples.

As shown in Figure 3(a1), control samples (full and empty green symbols) are grouped in the negative part of the first Principal Component (PC) and are well discriminated by OC cases (full and empty red symbols), characterized by higher PC1 scores and more dispersed along with the first and second PCs. Notably, the overlap between these two groups is very limited. Benign disease samples (full and empty yellow symbols), located in the middle of the plot, show partial overlap with both cases and controls. No grouping can be highlighted considering the pre- (empty symbols) and post (full symbols) menopausal patients.

In Figure 3(a2), early-stage patients are put in evidence (empty black symbols) to evaluate their distribution in the plot. Early-stage patients are dispersed in the right side of the plot along with cases, thus highlighting the ability of the e-nose to recognize them as cases and to discriminate early stages from healthy controls and benign. The loading plot (Figure 3b) identified four sensors, which were more suitable to discriminate among different samples: S1 sensor was selected for its ability to characterize control samples; S2, S6, and S7 sensors were selected for their ability to discriminate OC cases through significantly increased response signals. The ability of the e-nose in identifying OC cases was irrespective from the CA125 levels (data not shown) and from the tumor size (≤3/>3 cm) (Appendix A). The e-nose was able to discriminate OC cases based on their histology (high-grade serous vs. other). Although all OC cases were clustered in the same group, cases characterized by non-serous histology were homogeneously grouped on one corner of the OC group, thus suggesting that e-nose could potentially discriminate among different histology and biology (Appendix A).

### 2.2. K-Nearest Neighbours Models Development

The predefined groups were used as a-priori information (Y) to build classification models based on the e-nose data collected by the four selected sensors (X). The applied classifier (KNN) was evaluated by three metrics: sensitivity, specificity, and accuracy. These were computed on the basis of four factors (Appendix A): True Positive (TP), False Positive (FP), True Negative (TN), and False Negative (FN). Two models were elaborated to discriminate between OC cases and non-cancer patients. Both models were developed using strict and most probable prediction approaches for a class assignment. A first 1- K-NN (cases vs controls) model was developed to discriminate between OC cases and controls (Table 1).

The model was built using a calibration dataset of 58 cases and 76 controls, which was internally validated (cross-validation). The sensitivity, specificity, and accuracy of the strict class prediction approach were 95%, 93%, and 95%, respectively. Two out of the nine misclassified patients were early-stage patients, while the other 21 early-stage patients included in the cross-validation were correctly recognized as OC cases.

The remaining e-nose data (28 cases and 39 controls) were used to test the model capability in prediction, and confirmed its reliability to discriminate cases from controls (98% of sensitivity, 95% of specificity, and 96% of accuracy). Only a single early-stage patient was misclassified as a control, while the other eight early-stage patients included in the prediction test set were correctly assigned to OC cases. The most probable class assignation gave a perfect prediction of both classes, with a 100% accuracy.

A second 1-K-NN (cases vs. controls + benign) model was built by grouping the two non-cancer groups (benign disease and controls). The two classes considered in this model were: OC cases (*n* = 86) and controls (*n* = 114) plus benign disease (*n* = 51). The original dataset was split into a calibration set (70% of the data) and a test set (30% of the data). The model performance was tested in cross-validation and prediction. When the class prediction was applied, a sensitivity of 86%, a specificity of 84%, and an accuracy of 85% were achieved in cross-validation. Four out of 23 early-stage patients and two out of 14 patients with tumor size ≤ 3 cm were misclassified in the cross-validation phase. The prediction’s performance of this model had 89% sensitivity, 86% specificity, and 87% accuracy. In this validation phase, only one out of nine early-stage patients was misclassified. The most probable class assignation gave a perfect prediction of both classes, with a 100% accuracy (Table 2).

## 3. Discussion

The present study investigated the role of e-nose as a diagnostic tool for the detection of OC, thus reporting several noteworthy findings. First, e-nose could discriminate between breath samples collected from OC cases and non-cancer patients, albeit in a small and limited cohort of patients. Second, our study identifies four specific e-nose sensors, which can have a potentially relevant role in the detection of OC cases. Third, our models are characterized by high levels of sensitivity, specificity, and accuracy for the analyzed cohort of samples. Additionally, we observed that e-nose could discriminate between early-stage and advanced OC and controls, also regardless of patients’ age. However, this observation was limited due to the small numbers, unequal distribution of cancer subtypes, and the low prevalence of the early-stage disease in our cohort.

The exhaled breath contains hundreds of VOCs, which reflect the metabolic processes of both normal and cancer cells, and can be detected by the e-nose technology. The feasibility of the e-nose in the diagnostic workup of several diseases has been investigated, suggesting that potentially e-nose might discriminate among types of different diseases [12,13,14,15,16,17,18,19,20,21]. There is a growing body of literature suggesting that e-nose could be considered a useful tool for the early detection of lung and head and neck cancers [15,19,20,23]. A recently published review and meta-analysis on 14 studies testing the e-nose in patients with lung cancer, suggested that this method has high (>80%) sensitivity and specificity in discriminating between patients with lung cancer and healthy subjects [9]. In the field of oncology, e-nose might be useful not only from a diagnostic point of view but also for monitoring disease response during treatment [24]. Recently, de Vries observed that the e-nose assessment was effective for the non-invasive prediction of a patient’s disease response during immunotherapy [24]. In a prospective study on 143 non-small cell lung cancer patients, e-nose allowed baseline discrimination between responders and non-responders to immune checkpoint inhibitors, with potential implications in predicting response to treatment [24].

An in vitro study tested e-nose for the detection of OC [25]. This study compared samples of high-grade serous OC with samples of healthy tissues (adnexal and uterine tissues). They observed that the e-nose correctly classified 84.4% and 86.8% of cancer and healthy tissues, respectively [25]. More recently, Amal et al. performed a pilot study testing the role of e-nose in OC prediction [26]. This study analyzed samples and data of 182 women, with the following distribution among groups: 48 OC patients, 48 controls, and 86 patients with benign disease. The nanoarray output showed the potential utility of e-nose in discriminating between OC patients and controls (79% sensitivity, 100% specificity, and 89% accuracy) [26]. These preliminary results highlighted that efforts are needed to improve the sensitivity and accuracy of the e-nose. Here, we developed two models that could improve the diagnostic value of e-nose.

Screening and diagnostic methods for early diagnosis of OC represent an unmet clinical need, as no reliable screening and diagnostic devices are available. Since the diagnosis of OC is challenging, investigation of new a highly sensitive (at least 80%) and specific (as close as possible to 100%) test is paramount. We auspicate that the adoption of e-nose (in addition to conventional tests) would be useful in triaging patients needing further diagnostic and therapeutic procedures. Based on the results of the present exploratory study, the proposed approach seems to be promising; but it needs further validation. Besides, even in the presence of benign gynecologic disease, the 1K-NN model approximates quite satisfactorily the target values established for specificity and sensitivity when a strict prediction approach was applied, while for the most probable prediction approach, the target values are satisfied.

In the present paper, we performed a preliminary analysis evaluating the use of the e-nose in detecting OC. Our paper highlighted that the specificity in discriminating between OC and benign lesions was 86%. Although specificity is not very high, it is similar to those achieved with other available models. In particular, the IOTA group (International Ovarian Tumor Analysis) elaborated models to identify early-stage OC. In a recent paper, the IOTA group compared the performance of the Simple Rules Risk (SRR), and the Assessment of Different NEoplasias in the adneXa (ADNEX) model. This model was based on ultrasound, CA125 levels, and several other anamnestic features. At a 1% risk cut-off, specificity for SRR was 38.0% (35.5% to 40.6%), and for ADNEX was 19.4% (17.4% to 21.5%). At a 30% risk cut-off, specificity for SRR was 81.1% (79% to 83%), and for ADNEX was 84.5% (82.6% to 86.3%). The specificity reported with e-nose is higher than the specificity provided with these models, which is quite low [27]. Moreover, as evidenced in Table 2, the most probable class assignation gave a perfect prediction of both classes leading to 100% sensitivity and specificity.

E-nose achieved higher specificity values than those obtained based on a combination of CA125 levels and ultrasound. It is not surprising, since ultrasound and CA125 are not characterized with high specificity. Moreover, we can speculate that a new diagnostic algorithm, incorporating e-nose data with anamnestic, clinical, radiological (i.e., ultrasound) and humoral (i.e., CA125) features, would improve the specificity of this test that allows us to identify patients with OC in the early phase of the disease. The main strength of the present investigation includes the prospective evaluation of consecutive women diagnosed with suspected ovarian masses (including OC and benign ovarian masses) and healthy subjects. Other strengths of the study are: (i) the possible application of a simplified e-nose (based on the four selected sensors) that represents an innovative and promising tool for screening, diagnosis, and prognosis of patients with OC; and (ii) the development of two mathematical models that might be applied to simplify the interpretation of the results provided by e-nose.

Six points of our study deserve to be addressed. First, we observed that altered VOCs is a symptom of OC; therefore, the next step will be to investigate in a large population if altered VOCs might be predictive for higher risk of developing OC in comparison to normal VOCs, allowing to identify patients at risk, regardless the presence of ovarian masses. Second, the presence of different types of OC may have influenced the findings of the current study; future studies should stratify the results according to disease histology to assess potential differences in the e-nose detection accuracy. Third, the impact of other tumors (e.g., lung cancer) on the VOCs has to be tested to investigate the potential confounding effects on VOCs. Fourth, the inherent biases related to the relatively small sample size requires further studies to confirm the clear efficacy profile of e-nose. Fifth, further evidence is needed to assess the role of e-nose for screening and diagnostic purposes. Finally, our study suggested that the size of the tumor (<3cm vs. >3 cm) did not impact on the diagnostic ability of e-nose. However, further investigations are needed to assess the possible existence of the “lower detectable volume” and to clarify the effective diagnostic value of e-nose in the early phase of the disease. It is important to highlight that our results should be considered preliminary. This is a preliminary experience to determine whether this technology could successfully triage patients with OC. Further evidence is warranted to evaluate the real advantages of adopting e-nose and how to improve the specificity of this approach.

## 4. Methods

### 4.1. Ethical Approval and Recruitment of Subjects

This is a case-control prospective study. The Institutional Review Board (IRB) approved this study (INT-D237354: 011/11). The study population included all consecutive patients with adnexal masses treated at the National Cancer Institute (Milan - Italy) from 1 March 2018 to 30 April 2019. All the patients signed written consent for research purposes. The exclusion criteria were: age < 18 years, presence of synchronous solid cancer (within five years), surgery for recurrent disease, and consent withdrawn. All consecutive patients diagnosed with adnexal masses scheduled to have surgery were included in the study. Additionally, we included a control group of healthy subjects without ovarian abnormalities. All the controls had a negative ultrasonographic examination. According to the study protocol, all exhaled breaths were analyzed on the same day of sample collection. OC patients included in the study had staging surgery or cytoreduction according to the disease characteristics. Details regarding our surgical practice were reported elsewhere [28,29]. The stage of the disease and grading were assessed using the International Federation of Obstetrics and Gynecologists (FIGO) system. Histological subtypes were reported according to the World Health Organization (WHO) system [29]. Patients with adnexal masses were tested the day before surgery, while healthy controls were tested at their convenience. No patient was under active oncologic treatments. The present research follows both the STARD and Transparent Reporting of a multivariable prediction model for Individual Prognosis or Diagnosis (TRIPOD) guidelines for a study investigating diagnostic accuracy [30,31]. The objective of the adoption of the STARD and TRIPOD guidelines is to improve the completeness and transparency of the reporting of studies of diagnostic accuracy, to allow readers to assess the potential for bias in the study (internal validity), and to evaluate its generalizability (external validity) [30,31].

### 4.2. Sample Collection

The sample collection started without specific conditions since 2016. A preliminary feasibility study was conducted collecting data from 150 patients. The evaluation of these preliminary data allowed the identification of the correct sampling condition and its reliability (data regarding our preliminary experience are reported in Appendix A). For the present work, e-nose data were collected from 1 March 2018, to 30 April 2019. During this period, exhaled breath samples were collected between 7.00 and 7.30 a.m. from fasting patients (i.e., patients were not allowed to take any food and beverages, except water, since midnight before sampling. Similarly, smoking was not allowed since the night before sampling. An accurate description of the method used for the collection of exhaled breath is reported elsewhere [17]. Briefly, subjects breathed through a Teflon mouthpiece in a homemade Nalophan bag (30 × 20 cm) taking care to evaluate alveolar breath that refers to the last portion of the exhaled breath, expelled from within the lungs and the lower-airways, which have undergone gaseous exchange with the blood in the alveoli. A healthy adult expires approximately 500 mL air with each breath, of which the first 150 mL consists of dead-space air (no transfer of oxygen) from the upper air ways and nasopharynx. Therefore, patients and controls were asked to perform a single slow vital capacity breath, to trap the last 350 mL of exhaled breath. Breath samples were analyzed by e-nose on the same collection day.

### 4.3. Electronic Nose Analysis

Measurements of breath samples were performed by a commercial and portable e-nose (PEN 3, Win Muster Airsens Analytic Inc., Schwerin, Germany). The sensor array is composed of 10 Metal Oxide Semiconductor (MOS) sensors of different chemical composition, thickness and working temperature, to provide sensitivity and selectivity towards volatile compounds as indicated by the instrument supplier: S1 (aromatic compounds), S2 (broad-range compounds, polar compounds, nitrogen oxides, and ozone), S3 (ammonia, aromatic compounds, aldehydes, and ketones), S4 (hydrogen), S5 (alkanes, aromatic compounds, and less polar compounds), S6 (methane and broad-range compounds), S7 (sulfur compounds, terpenes, and sulfur organic compounds), S8 (alcohols, partially aromatic compounds, and ketones), S9 (aromatic compounds and sulfur organic compounds), and S10 (methane). During the sampling procedure, the breath samples contained in the bags were pumped over the sensor surfaces for 50 seconds (sampling time) during which the sensor responses were recorded; the flow rate was set at 400 mL/minutes. After sample analysis, the sensors were purged for 180 seconds with filtered air (purging time), then, before the next sample injection, the sensor baselines were re-established for 5 seconds. The sensor response corresponded to the fractional value obtained by subtracting the resistance signal of the baseline (R0-ohm) from the resistance signal of the sensors (R-ohm) and dividing by the resistance signal of the baseline (R0), thus providing a dimensionless, normalized response. Data were acquired after 40 seconds of sampling. Each breath sample was analyzed in duplicate, and the average of the sensor data was used for the statistical elaboration. Biostatisticians who analyzed e-nose data were blinded (i.e., not aware of the patients’ diagnoses) when performing the analyses.

### 4.4. Statistical Analysis

The e-nose data were transformed by column autoscaling and then explored by basic descriptive statistics and by multivariate statistical techniques as Principal Component Analysis (PCA) and the K-Nearest Neighbors’ algorithm (K-NN). PCA is an unsupervised exploratory procedure that allows us to explore in a reduced space the data structure and the relationships between objects and variables, thanks to graphical outputs (i.e., score plot, loading plot, and bi-plot) [32,33]. To develop the classification models, the K-Nearest Neighbors’ algorithm (K-NN) was used. K-NN is a simple non-linear classification approach based on Euclidean distance; it is one of the less prone to overfitting thanks to the simplicity of its algorithm, which does not include the optimization of numerous parameters and does not require any assumptions on the underlying data distribution. In detail, K-NN predicts the class membership of a sample based on the class of the K nearest sample(s) in the multidimensional space; in this work, a K value of 1 was applied. Before the development of the classification models, each dataset was split by the Kennard–Stone algorithm [32,33,34] into a training and a test set composed of about 70% and 30% of the original samples, respectively. The training dataset was used for the classification rule development, and the internal validation by cross-validation with five cancellation groups, whereas the test set was used to test the model’s performance in the prediction. The class predictions are based on predicted class probabilities for samples; different approaches can be applied to assign a sample to a specific class. In this work we applied two strategies, namely strict class prediction and most probable prediction. Strict class prediction assigns each sample to a specific class if the belonging probability is greater than a specified threshold probability value (*p* > 0.5) for one and only one class. This approach leads to the possibility of unassigned or multiple assignments of the samples. Indeed, if a sample has a probability lower than the threshold for the two classes, no class could be assigned to the sample. On the contrary, if a sample exceeds the threshold in the two considered classes, it will be assigned to both. This prediction provides the most safety in-class assignment but could lead to poorer figures of merits. Moreover, the most probable approach has been applied to assign samples to classes. In this case, samples are assigned to the class reaching the highest probability regardless of the magnitude of that probability. Even if more than one class has a probability higher than 0.50, the sample will be assigned to the one with the highest probability; likewise, if all probabilities are below 0.50, the sample will be assigned to the one with the largest probability. This approach is advantageous if there is a need for a class assignment and if "no class" or “multiple class” has no meaning. In our case, this approach is adequate since we must assign samples in the predefined classes.

For both approaches, the classification performance was evaluated by considering the accuracy, corresponding to the probability of doing a correct classification; the sensitivity, corresponding to the probability to classify a subject as a case when this is true (true positive rate); the specificity, corresponding to the probability of classifying a subject as control when this is true (true negative rate). All the data analyses were performed under the Matlab environment (R2017b, The Mathworks Inc., Natick, MA, USA), eventually using the PLS toolbox (ver. 8.5, Eigenvector Research Inc., Manson, WA, USA) software package.

## 5. Conclusions

Our data suggest that e-nose is a non-invasive, appealing technique that might be useful during an OC workup. To date, our data should be considered preliminary. They represent the basis for further studies. Further large prospective studies are needed to confirm our data. OC represents a group of heterogeneous diseases [35,36,37,38,39,40]; therefore, further investigations are needed to evaluate the applicability of e-nose in detecting various OC types. The new perspective is the implementation of this device in the new collection of exhaled breath samples on a larger population to validate these results and build more robust classification models. Additionally, further investigations would evaluate the role of e-nose alone and in combination with other diagnostic tools, such as the ROMA and ADNEX models.

## Figures and Tables

**Figure 1 cancers-12-02408-f001:**
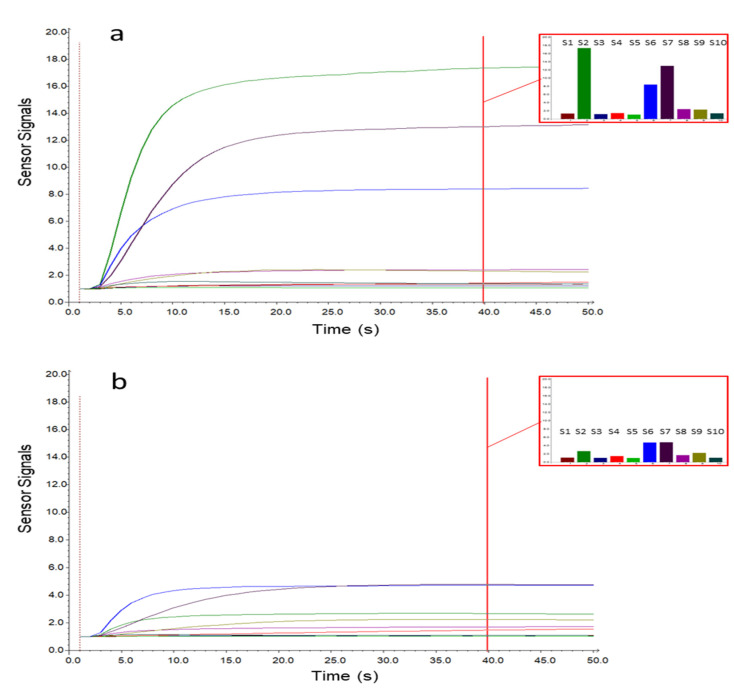
Metal oxide semiconductor (MOS) sensor signals for a case (**a**) and control (**b**) sample. The histograms represent the responses after 40 s of sampling for the 10 e-nose sensors (S1–10).

**Figure 2 cancers-12-02408-f002:**
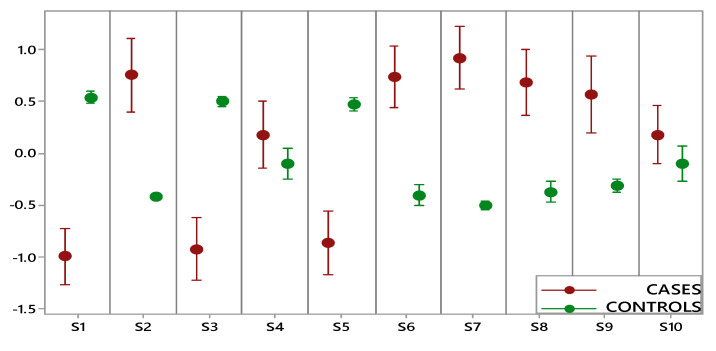
Interval plot of electronic nose data set for the ovarian cancer (OC) cases (red) and controls (green).

**Figure 3 cancers-12-02408-f003:**
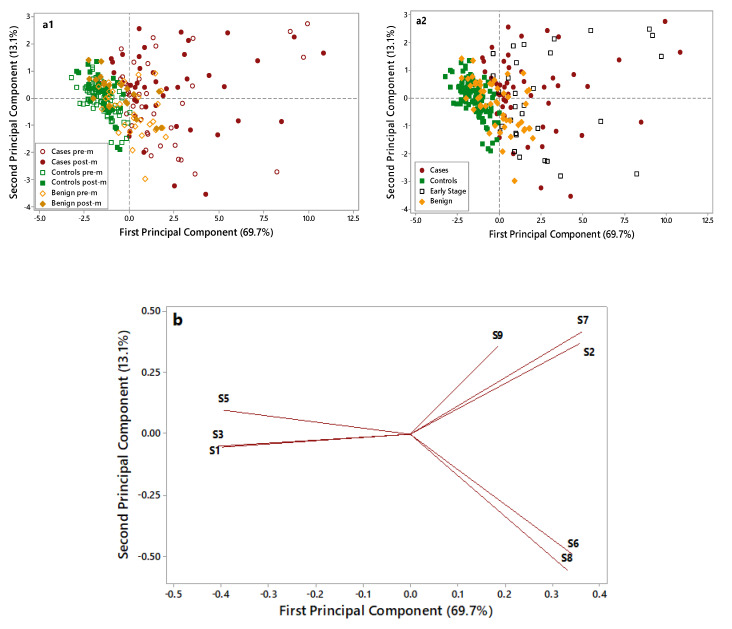
Principal Component Analysis(PCA) score plots (**a1**,**a2**) and loading plot (**b**) of electronic nose data collected on control (green), benign (yellow), and case (red) samples. In the score plot (**a1**) empty symbols are for premenopausal (pre-m) patients, and full symbols are for postmenopausal (post-m) patients. In the score plot (**a2**) empty black symbols are for early-stage patients. Two score pots were elaborated to: i. discriminate premenopausal and postmenopausal patients (Figure 3(a1)). ii. evaluate the plot distribution for early-stage ovarian cancer (OC) patients (Figure 3(a2)).

**Table 1 cancers-12-02408-t001:** Figures of merit for the K-Nearest Neighbors (KNN) model developed for cases vs. controls: sensitivity (Sens), specificity (Spec), and accuracy (Acc) are reported for the strict and the most probable assignation of the classes.

Model	Calibration					Cross-Validation			Prediction			
	Classes	*N*	Sens	Spec	Acc	Sens	Spec	Acc	*N*	Sens	Spec	Acc
Strict prediction	Cases	58	91%	95%	93%	93%	99%	95%	28	93%	100%	96%
	Controls	76	95%	91%	93%	96%	90%	95%	38	100%	93%	96%
	Global	134	94%	93%	93%	95%	93%	95%	66	98%	95%	96%
Most probable prediction	Cases	58	100%	100%	100%	93%	96%	95%	28	100%	100%	100%
	Controls	76	100%	100%	100%	96%	93%	95%	38	100%	100%	100%
	Global	134	100%	100%	100%	95%	94%	95%	66	100%	100%	100%

**Table 2 cancers-12-02408-t002:** Figures of merit for the KNN model developed for cases vs. controls + benign: sensitivity (Sens), specificity (Spec), and accuracy (Acc) are reported for the strict and the most probable assignation of the classes.

Model	Population	Calibration				Cross-Validation			Prediction			
	Classes	*N*	Sens	Spec	Acc	Sens	Spec	Acc	*N*	Sens	Spec	Acc
Strict prediction	cases	58	79%	89%	84%	81%	89%	85%	28	82%	93%	87%
	controls	110	89%	79%	84%	89%	81%	85%	55	93%	82%	87%
	global	168	86%	83%	84%	86%	84%	85%	83	89%	86%	87%
Most probable prediction	cases	58	100%	100%	100%	81%	89%	85%	28	100%	100%	100%
	Controls + benign	110	100%	100%	100%	89%	81%	85%	55	100%	100%	100%
	global	168	100%	100%	100%	86%	84%	85%	83	100%	100%	100%

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
