# Peer review of "Detection of Ovarian Cancer through Exhaled Breath by Electronic Nose: A Prospective Study"

_cancers, 2020, doi:10.3390/cancers12092408_

Round 1

Reviewer 1 Report

All earlier concerns were adequately addressed. The manuscript can be published as is.

Reviewer 2 Report

The authors have adequately addressed the raised questions

and I recommend the paper for publication.

This manuscript is a resubmission of an earlier submission. The following is a list of the peer review reports and author responses from that submission.

Round 1

Reviewer 1 Report

The paper is a feasibility study of volatile organic compounds in the expiration air as a new tool in the workup of patients suspected of having ovarian cancer.

The study deals with a new, interesting research issue. The detection of a disease in the expiration air (electronic nose) holds interesting perspectives from both a scientific and a clinical view. The study includes both patients with ovarian cancer and benign diseases together with a healthy control group. The cancer group represents different histopathologic types and clinical stages.

The results suggest a high sensitivity and specificity but also a high accuracy. It is interesting that the algorithm can be based on only 4 e-nose sensors.

The main limitations are the small number of patients in each group and the lack of independent validation. Thus, it is clearly a preliminary feasibility study as also stated by the authors.

In conclusion, the study addresses a new scientific issue and holds a quality that merits publication.

Author Response

The paper is a feasibility study of volatile organic compounds in the expiration air as a new tool in the workup of patients suspected of having ovarian cancer.

The study deals with a new, interesting research issue. The detection of a disease in the expiration air (electronic nose) holds interesting perspectives from both a scientific and a clinical view. The study includes both patients with ovarian cancer and benign diseases together with a healthy control group. The cancer group represents different histopathologic types and clinical stages.

The results suggest a high sensitivity and specificity but also a high accuracy. It is interesting that the algorithm can be based on only 4 e-nose sensors.

The main limitations are the small number of patients in each group and the lack of independent validation. Thus, it is clearly a preliminary feasibility study as also stated by the authors.

In conclusion, the study addresses a new scientific issue and holds a quality that merits publication

Answer: We thank the reviewer for these comments. No changes are requested

Reviewer 2 Report

The authors examine the potential of electronic odor sensor detection for the detection of ovarian cancer. Although novel means of early ovarian cancer detection could indeed have a major impact of the morbidity and mortality associated with this disease, the results presented are underwhelming. Although the authors acknowledge that this is a preliminary investigation, the idea of using electronic odor sensors for disease detection including the detection of cancer is not novel. Enthusiasm is further decreased due to the following specific points:

  1. The specificity of the proposed test to distinguish malignant lesions is 86%, which is such too low to make this clinically useful. This is likely to be even lower for patients with stage 1a tumors, raising serious doubts about the potential impact and about the likelihood that this approach could even outperform that of transvaginal ultrasound coupled with CA125 measurements.
  2. The authors used some patients with stage 1 tumors in their study population, but provided no information about tumor volume or substage; were there any stage 1a in their study cohort? What was the smallest tumor that gave a positive test? If all cases that tested positive had either large tumor volumes of had positive peritoneal cytology, then the potential impact is likely to be small. Correlation with CA125 levels would also be relevant.
  3. Which stages or tumor subtypes were used in Figs. 1 and 2?
  4. The authors conclude that the proposed technology could be useful in the evaluation of patients with suspected ovarian cancer. In which clinical context do they envision using this technology? Would it be as a general screening technique or in any other context? Also, would this technology be used as a general cancer screening tool or would it be applied specifically for screening of ovarian cancer? If the latter, why would this technology distinguish ovarian cancers from other cancers? 
  5. Although most of the cases had high grade serous tumors, the authors included other subtypes of ovarian cancer such as mucinous tumors, carcinosarcomas, and others. These different tumor subtypes have fundamentally different biologies. Were there any differences regarding the performance of the high-grade serous and other subtypes?

Author Response

All responses are attached 

Reviewer 3 Report

This is an interesting and well presented paper about an original and promising topic. The limits of the study have been adequately described. The methodology is accurate and adapted.

Some comments

The authors included a majority of pts with AOC rather than early stage, which is useful for the validation of their tool, but precludes any definitive conclusion regarding stages I-II. However, that is the main challenge for screening and early diagnosis. Further studies are necessary. The present one is preliminary.

For the diagnosis of adnexal masses, it could be useful to test e-nose in combination with imaging data, and also with the ROMA score. What would be the added value, if any ? (to be discussed)

L. 186 - 87 : the assertion about the prognostic value is not adequately supported. The reference cited concerns the prediction of treatment efficacy, not the prognosis.

L. 198 : the term "benign ... neoplasia" is incorrect.

L. 222 - 229 : some typos need to be corrected.

Author Response

Reviewer 3:

This is an interesting and well presented paper about an original and promising topic. The limits of the study have been adequately described. The methodology is accurate and adapted.

Some comments

Comment 1: The authors included a majority of pts with AOC rather than early stage, which is useful for the validation of their tool, but precludes any definitive conclusion regarding stages I-II. However, that is the main challenge for screening and early diagnosis. Further studies are necessary. The present one is preliminary.

Answer: In order to comply with the reviewer’s comment we highlighted this as a limitation of the present study.

Comment 2: For the diagnosis of adnexal masses, it could be useful to test e-nose in combination with imaging data, and also with the ROMA score. What would be the added value, if any ? (to be discussed)

Answer: We agree with the thoughts of the reviewer. Further investigations might evaluate the role of e-nose alone and in combination with other diagnostic tools, such as ROMA and ADNEX model. We point out this point in the discussion section.

Comment 3: L. 186 - 87 : the assertion about the prognostic value is not adequately supported. The reference cited concerns the prediction of treatment efficacy, not the prognosis.

Answer: In order to comply with the reviewer’s comment we modified this sentence “In the field of oncology, e-nose might be useful not only from a diagnostic point of view but also predicting treatments’ efficacy [24].”

Comment 4 : L. 198 : the term "benign ... neoplasia" is incorrect.

Answer: In order to comply with the reviewer’s comment we modified this sentence  “benign disease”

Comment 5: L. 222 - 229 : some typos need to be corrected.

Answer: In order to comply with the reviewer’s comment we checked and corrected the whole manuscript

Round 2

Reviewer 2 Report

I still have concerns in spite of the author’s replies to my earlier comments. This being a feasibility study, the goal is to demonstrate that this technology is promising enough to warrant further investigation. I am not convinced that the authors achieved this goal.

In response to the concerns about low specificity (previous comment #1), the authors provide data based on class assignment using “highest probability”, which is not stringent. I addition, the authors argue that they achieve specificity values that are higher than those obtained based on a combination of CA125 levels and ultrasound. However, they fail to mention that the latter approach is now regarded as of low merit precisely because of its low specificity. They should at least have some comments on their strategy to improve on specificity in order to support their argument that this approach needs to be further investigated.

Regarding my previous comments #2 and #3 inquiring about the lower detectable volume and disease stage, the authors gave vague answers including range of volumes in their study population as well as data suggesting that there is no correlation with size.  However, some insights about lower limit of detection and related metrics are precisely the kind of data needed in order to determine the potential of the proposed approach and whether or not further investigations are warranted. The information provided is not helpful in making this assessment.

My previous comment #4 asked for insights into the clinical context in which the authors anticipate that this technology will be useful. The authors avoid answering and simply state that such comments are not necessary for a feasibility study. It is very troublesome that the authors are not able to even speculate as to how this could be used clinically. Based on the current data presented in the manuscript, it is very unlikely that this approach will be specific enough to be clinically useful for early ovarian cancer detection if a positive result would imply that the patient should undergo surgery. My enthusiasm could be higher if they pointed out other scenarios where the technology would be useful or, at the very least, how it could complement other existing technologies.

Author Response

Comment 1: In response to the concerns about low specificity (previous comment #1), the authors provide data based on class assignment using “highest probability”, which is not stringent. I addition, the authors argue that they achieve specificity values that are higher than those obtained based on a combination of CA125 levels and ultrasound. However, they fail to mention that the latter approach is now regarded as of low merit precisely because of its low specificity. They should at least have some comments on their strategy to improve on specificity in order to support their argument that this approach needs to be further investigated.

Answer: Thank you for your comment. We added some more information in the text to better explain the peculiarity of the most probable prediction strategy. Indeed, the use of this predictor is advisable when each sample should belong to a single class., i.e. when "no class" or “multiple class” assignation has no meaning. In our case, this approach is adequate since we MUST assign each sample to one of the two predefined classes, thus “no class” or “multiple class” assignment has no meaning. This information has been added in the Material and Methods section (L353-355).

Furthermore, since the most probable class assignation leads to 100% of specificity, in our opinion it seems adequate also for future model development; considering, in any case, the need of larger sample numbers and the possibility of combining e-nose with other diagnostic tools.  This information has been added in the Conclusion section (L373-375).

Comment 2: Regarding my previous comments #2 and #3 inquiring about the lower detectable volume and disease stage, the authors gave vague answers including range of volumes in their study population as well as data suggesting that there is no correlation with size.  However, some insights about lower limit of detection and related metrics are precisely the kind of data needed in order to determine the potential of the proposed approach and whether or not further investigations are warranted. The information provided is not helpful in making this assessment.

Answer: The comment of the reviewer is very clever and interesting. In order to comply with his/her comment, we performed a PCA, identifying samples according to tumor size (≤3/>3 cm). Considering the score plot,  no sample distribution or clustering has been observed according to the tumor size. The PCA score plot was added as a supplemental figure (supplementary figure 1).

Moreover, considering the classification model (Strict prediction) built with cases vs controls+benign, it was possible to observe that 2 out of 14 patients with tumor size ≤ 3 cm were misclassified (this information has been added in the text line 170). Indeed, the percentage of correct classification was 86% in agreement with the predictive capability of the model (89% for sensitivity, 86% for specificity, and 87% for accuracy). Based on these findings we observed that the ability e-nose in identifying OC cases is independent of tumor size, thus it is not possible to identify the “lower detectable volume”. Moreover, we point out this as a possible limitation of our investigation. We suggested that further studies are needed to clarify this point: “.., our study suggested that the size of the tumor did not impact on the diagnostic ability of e-nose. However, further investigations are needed to assess the possible existence of the “lower detectable volume”, thus clarify the effective diagnostic value of e-nose in the early phase of disease”

Comment 3: My previous comment #4 asked for insights into the clinical context in which the authors anticipate that this technology will be useful. The authors avoid answering and simply state that such comments are not necessary for a feasibility study. It is very troublesome that the authors are not able to even speculate as to how this could be used clinically. Based on the current data presented in the manuscript, it is very unlikely that this approach will be specific enough to be clinically useful for early ovarian cancer detection if a positive result would imply that the patient should undergo surgery. My enthusiasm could be higher if they pointed out other scenarios where the technology would be useful or, at the very least, how it could complement other existing technologies.

Answer: In order to comply with the reviewer’s comment we pointed out this in the discussion section. 

“Screening and diagnostic methods for early identification of OC represent an unmet clinical need, as no reliable screening and diagnostic devices are available. Since the diagnosis of OC is challenging, investigation of new a highly sensitive (at least 80%) and specific (as close as possible to 100%) screening test is paramount. Therefore, our investigation might seed further studies aimed to improve the early detection of OC using both conventional tests and e-nose. This should allow the discrimination of the tumor-free patients, who will not undergo further controls, whereas the patients resulting positive to the e-nose test would be oriented to further second-level screening tests, up to biopsy”….”Moreover, we can speculate that a new diagnostic algorithm, incorporating e-nose data with anamnestic, clinical, radiological (ie, ultrasound) and humoral (ie, CA125) features, would improve the specificity of this test that allows us to identify patients with OC in the early phase of the disease”

Round 3

Reviewer 2 Report

The authors made some effort to address previous concerns. I remain unconvinced as to why this study belongs to a general interest journal like Cancers. I am also unconvinced that the data justify further studies in the context of ovarian cancer. The idea of using olfactory signals from various sources including breath for cancer detection is not novel, nor is the use of e-nose to aid in olfactory signal detection. The only novel aspect is that the authors studied ovarian cancer. Thus, the claim that this study is a proof of principle is only justified in this context. However, support for the conclusion this is a promising approach for early ovarian cancer screening  is very underwhelming. In response to an earlier comment, the authors state that this approach “should allow the discrimination of the tumor-free patients, who will not undergo further controls, whereas the patients resulting positive to the e-nose test would be oriented to further second-level screening tests, up to biopsy”. However, biopsy of an adnexal mass requires open surgery, making this scenario clinically impractical unless the proposed test has a specificity of 99.6% or higher. In addition, this would only apply to patients in whom an adnexal mass is detectable and therefore no longer an early cancer. Patients with positive e-nose tests in whom an adnexal mass is not detected would need an exploratory laparotomy, further underscoring the need for a highly specific test. The data presented suggest that cancer specificity is substantially lower than 99% while the authors provide no information on how specificity could be improved. This study would fit better in a specialized journal instead of a general interest journal like Cancers.

  1. It is stated on line 267 that the inclusion of several histotypes may have influenced the findings. This only reinforces the concern that the results may not be specific for high-grade serous ovarian carcinoma, as they might be skewed if another cancer type is present. Inclusion of controls with cancers other than ovarian cancer such as lung cancer and others would have been useful.

  1. The data for benign lesions overlapped with those of normal controls and cancers in Fig. 3a1. Inability to distinguish between cancerous and benign lesions raises serious concerns about the potential clinical utility of this approach.

  1. It is not clear whether or not the patients with cancer were under treatment when breath samples were obtained. It is certainly possible that chemotherapy or even other forms of treatment could influence the results.

  1. The authors conclude that their technology can identify ovarian cancers independently of tumor volume based on a cut-off of 3cm for low versus high volumes. Most aggressive cancers have probably already spread when the primary tumor reached 3cm, meaning that the proposed approach needs to show positive results with much smaller tumors in order to be clinically useful.

  1. The data on specificity were substantially better when the “most probable approach” method was used to assign samples to classes. Were the biostatisticians aware of the diagnoses when making the assignments based on this approach. The merit of the data based on this approach is also difficult to evaluate because the authors are unclear about the details of their “most probable approach” protocol.

  1. The entire manuscript contains numerous typographical and grammatical errors, making it difficult to read and contributing to the impression of overall lack of attention to details. The following examples are taken exclusively from the introduction, which is only 35 lines, but are representative of the entire manuscript:

  1. The year 2019 is referred to as in the future on line 31 while data from that year are quoted as in the past on line 33
  2. Line 37: “Growing evidence suggested”
  3. Line 41: “lead to a non-negligible inaccuracies”
  4. Line 43: “might be a promising biomarkers
  5. The sentence from line 50 to 52 has no verb and is incomprehensible
  6. Line 56: “metabolome presents
  7. Line 53-54: “our study aim to identify”
  8. Line 65: “thus potentially seeding further researches” is awkward English

Author Response

The authors made some effort to address previous concerns. I remain unconvinced as to why this study belongs to a general interest journal like Cancers. I am also unconvinced that the data justify further studies in the context of ovarian cancer. The idea of using olfactory signals from various sources including breath for cancer detection is not novel, nor is the use of e-nose to aid in olfactory signal detection. The only novel aspect is that the authors studied ovarian cancer. Thus, the claim that this study is a proof of principle is only justified in this context. However, support for the conclusion is a promising approach for early ovarian cancer screening is very underwhelming. In response to an earlier comment, the authors state that this approach “should allow the discrimination of the tumor-free patients, who will not undergo further controls, whereas the patients resulting positive to the e-nose test would be oriented to further second-level screening tests, up to biopsy”. However, a biopsy of an adnexal mass requires open surgery, making this scenario clinically impractical unless the proposed test has a specificity of 99.6% or higher. In addition, this would only apply to patients in whom an adnexal mass is detectable and therefore no longer early cancer. Patients with positive e-nose tests in whom an adnexal mass is not detected would need an exploratory laparotomy, further underscoring the need for a highly specific test. The data presented suggest that cancer specificity is substantially lower than 99% while the authors provide no information on how specificity could be improved. This study would fit better in a specialized journal instead of a general-interest journal like Cancers.

We replied to all comments requested by the authors for round 1 and 2. Similarly, we addressed all comments for round 3. To comply with the reviewer’s comment we limit over the interpretation of our results. We highlighted that our study is a preliminary experience and further larger studies are warranted. Further experience would be useful to address the real advantages of adopting e-nose and how to improve and specificity of this approach. Further investigations would evaluate the role of e-nose alone and in combination with other diagnostic tools, such as ROMA and ADNEX model. The paper was sent to a special issue titled: Diagnostics, Staging, and Surgical Treatment of Gynaecological Cancer. 

Specific Comment 

Comment 1: It is stated in line 267 that the inclusion of several histotypes may have influenced the findings. This only reinforces the concern that the results may not be specific for high-grade serous ovarian carcinoma, as they might be skewed if another cancer type is present. Inclusion of controls with cancers other than ovarian cancer such as lung cancer and others would have been useful.

Answer: As previously reported in our paper, this is a preliminary experience looking at the feasibility of e-nose to identify patients with OC. We addressed all these concerns in the discussion section

“Six points of our study deserve to be addressed. First, in the present study we observed that altered VOCs is a symptom of OC; therefore, the next step will be to investigate in a large population if altered VOCs might be predictive for higher risk of developing OC in comparison to normal VOCs, thus allowing to identify patients at risk, regardless the presence of ovarian masses. Second, the presence of different types of OC may have influenced the findings of the current study; thus, future studies should stratify the results according to the histotype to assess potential differences in the accuracy of detection by the e-nose. Third, the impact of various diseases other than OC (for instance lung cancer) on VOCs has to be tested, to investigate the potential confounding effects on VOCs. Fourth, the inherent biases related to the relatively small sample size requires further studies to confirm the clear efficacy profile of e-nose. Fifth, further evidence is needed to assess the role of e-nose for screening and diagnostic purposes. Finally, our study suggested that the size of the tumor did not impact on the diagnostic ability of e-nose. However, further investigations are needed to assess the possible existence of the “lower detectable volume”, thus clarify the effective diagnostic value of e-nose in the early phase of the disease. It is important to highlight that our results should be considered preliminary. This is a preliminary experiment to determine whether this technology could successfully triage patients with OC. Further evidence is warranted to real advantages of adopting e-nose and how to improve and specificity of this approach”.  

Comment 2: The data for benign lesions overlapped with those of normal controls and cancers in Fig. 3a1. The inability to distinguish between cancerous and benign lesions raises serious concerns about the potential clinical utility of this approach.

As we stated in our paper PCA shows that benign lesions are partially overlapped with cases and controls. For this reason, we applied a statistical model that allows us to discriminate between benign and malignant lesions (please see K-Nearest Neighbours models development:).  

We addressed this point in our results section: 

“As can be seen in Fig. 3a1 control samples (full and empty green symbols) are grouped in the negative part of the first Principal Component (PC) well discriminated by OC cases (full and empty red symbols) characterized by higher PC1 scores and more dispersed along with the first and second PCs; the overlap between these two groups is very limited. Benign samples (full and empty yellow symbols), located in the middle of the plot, are partially overlapped with cases and controls. No grouping can be highlighted considering the pre (empty symbols) and post (full symbols) menopausal patients… 1-K-NN (cases vs controls + benign) model was built by grouping the two non-cancer groups (benign + controls), thus considering two new classes: OC cases (86) and controls + benign (114+51). The original data set was split into a calibration set (70% of the data) and a test set (30% of the data). The model performance was tested in cross-validation and prediction. When the class prediction was applied, a sensitivity of 86%, a specificity of 84%, and an accuracy of 85% were achieved in cross validation…. The performance recorded in prediction was 89% for sensitivity, 86% for specificity, and 87% for accuracy. In this validation phase, only one out of nine early-stage patients considered was misclassified. The most probable class assignation gave a perfect prediction of both classes leading to 100% accuracy (Table 2).”

Comment 3: It is not clear whether or not the patients with cancer were under treatment when breath samples were obtained. It is certainly possible that chemotherapy or even other forms of treatment could influence the results.

Answer: to comply with the reviewer’s comment we clarified this point. All patients were tested before surgery, as reported in the method section we evaluated patients with suspected lesions, no patients had the diagnosis of malignancies before e-nose testing. 

Comment 4: The authors conclude that their technology can identify ovarian cancers independently of tumor volume based on a cut-off of 3cm for low versus high volumes. Most aggressive cancers have probably already spread when the primary tumor reached 3cm, meaning that the proposed approach needs to show positive results with much smaller tumors to be clinically useful.

Answer: To comply with the reviewer’s comment we addressed this is the issue in the discussion section. As previously reported our study is a preliminary experience to assess the potential role of e-nose in identifying OC patients. 

our study suggested that the size of the tumor (<3cm vs >3 cm) did not impact on the diagnostic ability of e-nose. However, further investigations are needed to assess the possible existence of the “lower detectable volume”, thus clarify the effective diagnostic value of e-nose in the early phase of the disease.

Comment 5: The data on specificity was substantially better when the “most probable approach” method was used to assign samples to classes. Were the biostatisticians aware of the diagnoses when making the assignments based on this approach? The merit of the data based on this approach is also difficult to evaluate because the authors are unclear about the details of their “most probable approach” protocol.

Answer: all e-nose analyses were performed by the Department of Food, Environmental and Nutritional Sciences (DeFENS), Università degli Studi di Milano (the day before surgery). Biostatisticians were not aware of the diagnoses when they make the analyses. We clarified this point in the method section.

Comment 6: The entire manuscript contains numerous typographical and grammatical errors, making it difficult to read and contributing to the impression of overall lack of attention to details. The following examples are taken exclusively from the introduction, which is only 35 lines, but are representative of the entire manuscript:

1. The year 2019 is referred to as in the future on line 31 while data from that year are quoted as in the past on line 33

2. Line 37: “Growing evidence suggested”

3. Line 41: “lead to a non-negligible inaccuracies”

4. Line 43: “might be a promising biomarkers”

5. The sentence from line 50 to 52 has no verb and is incomprehensible

6. Line 56: “metabolome presents

7. Line 53-54: “our study aim to identify”

8. Line 65: “thus potentially seeding further researches” is awkward English

Answer: The language was revised by a native English speaker. All errors were corrected.